# Faster Evaluation of Dimensional Machine Performance in Additive Manufacturing by Using COMPAQT Parts

Laurent Spitaels [1,*], Endika Nieto Fuentes [2], Valentin Dambly [1], Edouard Rivière-Lorphèvre [1], Pedro-José Arrazola [2] and François Ducobu [1]

1  UMONS Research Institute for Materials Science and Engineering, University of Mons, Place du Parc 20, 7000 Mons, Belgium; valentin.dambly@umons.ac.be (V.D.); edouard.rivierelorphevre@umons.ac.be (E.R.-L.); francois.ducobu@umons.ac.be (F.D.)
2  Faculty of Engineering, Mondragon Unibertsitatea, Loramendi 4, 20500 Arrasate-Mondragón, Spain; endika.nieto@alumni.mondragon.edu (E.N.F.); pjarrazola@mondragon.edu (P.-J.A.)
*  Correspondence: laurent.spitaels@umons.ac.be

**Abstract:** Knowing the tolerance interval capabilities (TICs) of a manufacturing process is of prime interest, especially if specifications link the manufacturer to a customer. These TICs can be determined using the machine performance concept of ISO 22514. However, few works have applied this to Additive Manufacturing printers, while testing most of the printing area as recommended takes a very long time (nearly 1 month is common). This paper, by proposing a novel part design called COMPAQT (Component for Machine Performances Assessment in Quick Time), aims at giving the same level of printing area coverage, while keeping the manufacturing time below 24 h. The method was successfully tested on a material extrusion printer. It allowed the determination of potential and real machine tolerance interval capabilities. Independently of the feature size, those aligned with the X axis achieved lower TICs than those aligned with the Y axis, while the Z axis exhibited the best performance. The measurements specific to one part exhibited a systematic error centered around 0 mm $\pm$ 0.050 mm, while those involving two parts reached up to 0.314 mm of deviation. COMPAQT can be used in two applications: evaluating printer tolerance interval capabilities and tracking its long-term performance by incorporating it into batches of other parts.

**Keywords:** Additive Manufacturing; material extrusion; machine performance; capability; tolerance; COMPAQT

## 1. Introduction

Additive Manufacturing (AM) processes are paving the way to the production of complex designs, even for small batches or unique parts [1]. The possible materials are thermoplastics, composites, metals, and even ceramics [2]. However, due to their relative youth (AM processes first appeared in the 1980s [3]), these processes are still not yet fully mastered and require further investigations. Their current limitations encompass slow building rates, higher specific energy consumption, variable surface finish, and large dimensional and geometrical tolerances [4–7]. The relative inaccuracy of these processes and their inability to reach smooth surface topographies require carrying out finishing operations with mechanical, thermal, and chemical finishing processes prior to their use in complex mechanical assemblies, for example [8]. Nevertheless, the AM processes are seen as game changers for industry 4.0, and among the seven families of processes defined by ISO 52900 [9], Material Extrusion (MEX) is seen as one of the most promising in a 10-year horizon [10].

Tolerance intervals are related to design/part specifications and can be applied to any part built using AM or any other process. On the other hand, tolerance interval capabilities (TICs) are used to determine if a process can achieve the desired tolerance set by the function and specified using standards. Therefore, knowing the achievable tolerance

interval capabilities of AM printers is key before planning the finishing operations [11]. This is mandatory in order to predict the needed stock allowance for machining operations, for example. Standards for AM printers are progressively released by two working groups, ASTM F42 and ISO/TC 261 [12], while standardized methods for the determination of the achievable tolerances must still be developed. The achievable tolerances are also required when foreseeing the use of a 3D-printed part directly in an assembly and when interfacing it with other components [11]. The resulting possible fitting between the assembly parts is directly linked to the achievable tolerances. For the selection of dimensional tolerances, the part designer can rely on standards such as ISO 286-1 [13] for Standard Tolerance Grades (STGs from IT01 to IT18), while tables of geometrical tolerances can be found in ISO 2768-2 [14], for example. These standards are widely used for conventional processes (e.g., as machining). They can also be used for AM processes according to ISO 17296-3 [15] since no AM-dedicated standard already exists for this purpose.

Most of the existing methods for determining achievable tolerances rely on the printing and measurement of artifact parts. Usually, these parts, often called GBTAs (Geometrical Benchmark Test Artifacts), are replicated five times and then measured to obtain the short-term performances of the tested machine. Many works have already explored this method [16,17], while the recent ISO 52902 standard [18] tried to give a unified reference of the guidelines established through the literature. For example, the work of Lieneke et al. [19] determined the short-term performances of three printers using Material Extrusion and Powder Bed Fusion by replicating 10 mm edge cubes at different locations in each machine. The spatial repeatability of each printer was assessed, while a comparison of the printers' relative performances was performed, as well as a comparison with other conventional processes using the STG of ISO 286-1. The other GBTAs proposed in the literature include various geometries such as cubes, cylinders, slopes, and spheres in order to also allow the geometrical deviations to be evaluated [16]. However, the proposed GBTA parts require a lot of time to be printed since they aim to cover as much as possible the building platform of the printer (at least 80%) while requiring a thick base (more than 5 mm according to Minetola et al. [20]) in order to avoid their permanent deformation after removing them from the printer. Even if this approach leads to a systematic determination of the tolerance interval capabilities, it is time and material consuming. Indeed, the GBTA base requires a significant amount of material with respect to the features to be measured on its top. For example, the GBTA proposed in a previous study [21] used a base of 10 mm high and required 27 h in order to be printed. The functional features of the GBTA (stairs, planes, cylinders, and hemispheres) accounted for about 10% of the total deposited material to obtain one part, while the thick base accounted for 85% of the total printing time. Nevertheless, this part design allows for conducting within the same part a dimensional and geometrical performance analysis, while systematically covering the printer's achievable dimensional size ranges of ISO 286-1. Therefore, the GBTAs are well suited for R&D purposes and short-term performance evaluation, while their use in an industrial context is compromised due to the time and cost they require.

When foreseeing a long-term performance analysis of a machine, the concept of capability can be used [22]. The different indicators and methods to apply while assessing the capability of machines are defined by ISO 22514-3 [23]. Two main types of studies can be conducted: process capability and machine performance. Process capability considers the influence of the machine itself and external factors such as human, material, method, and environment [24]. However, this analysis can be performed only if process stability was demonstrated. In most of the literature presenting the performances assessment of AM printers, this condition is not fulfilled [25]. The analysis can then be limited to a machine performance study encompassing the influence of the machine without its external factors [24]. Still, even if external factors are not taken into account for the machine performance study, it encompasses the factors affecting the machine itself (the numerical chain from the CAD model to the slicing, for example). Therefore, the chord and angle deviation when exporting the CAD model in .STL can affect the machine performance

results. For the present paper, machine performance analysis was considered since the process stability of the selected printer was not yet demonstrated.

Several ranges of dimensions can be considered as those introduced by ISO 286-1 when referring to dimensional tolerances. Choosing a part design encompassing all the achievable size ranges is then of prime interest to best characterize its possibilities with respect to this standard. Moreover, the distribution of the deviations associated with a measurement (dimensional or geometrical) is not always best modeled by a normal distribution [23]. Testing different distributions to best fit the collected data is then mandatory to enable the computation of the required percentiles used in machine performance indices. Finally, performing a machine performance study with existing tolerance intervals, as those widely used in conventional processes (ISO 286-1 and ISO 2768-2 for dimensional and geometrical tolerances, respectively), enables a comparison between them and the tested AM printer. This enables a quantification on the same basis of the gain brought by finishing operations conducted on AM-printed parts. This is especially interesting when designing hybrid machines combining additive and subtractive processes (milling, for example).

The ISO 22514 standard method was successfully applied by some recent papers in the context of determining AM printer performances [22,25–28], with only one of them [25]:

- Encompassing all the dimensional size ranges of ISO 286-1 achievable by the tested printer for its X and Y axes;
- Identifying for each set of data the best-fitting distribution between the normal, log-normal, folded normal, gamma, Rayleigh, and Weibull distributions;
- Basing its analysis on the tolerance intervals from ISO 286-1 for the dimensional measurements and ISO 2768-2 for the geometrical measurements.

Even if that method was systematic, it required 675 h to produce the 25 GBTAs to be measured, making it impossible to use it in an industrial context where time and costs were aimed to be optimized. The other studies [22,26–28] relied on a simpler design than GBTA, and none mentioned the time needed to manufacture the parts, except the study of Udroiu et al. [27], which required about 6 h of printing to produce 150 parts (3 batches of 50) in the tested printer. Nevertheless, the parts manufactured for the analysis were produced without being distributed across the printing area. In addition, the design used did not allow for covering the printer's achievable dimensional size ranges of ISO 286-1.

Much research has been devoted to assessing the short-term performances of AM printers [16,17]. However, only few papers have tackled longer-term performance assessment using the ISO 22514 standard [22,25–28]. Among them, only one [25] systematically covered the printer's achievable dimensional size ranges of ISO 286-1. However, the printing time for obtaining the required number of parts (675 h) does not allow for foreseeing an industrial use. Therefore, the present paper aims to propose a method capable of delivering a machine performance analysis in line with ISO 22514, while keeping the production time of the parts lower than 24 h. The focus was set on dimensional measurements. Indeed, a part design encompassing various geometries such as cylinders, holes, and spheres entails a very high printing time, as each part produced must include different sizes of each geometry. The introduced COMPAQT (Component for Machine Performances Assessment in Quick Time) part design allows for systematically covering the ISO 286-1 dimensional size ranges from 1 mm to 180 mm for the printer's X and Y axes. The resulting method is suitable for production purposes and allows for performing a reception test (when commissioning a machine) or monitoring a longer-term production.

## 2. Method

### 2.1. Companion Part Design and Available Measurements

The design of the COMPAQT part is given in Figure 1a,b. This design was driven by the desire to best cover the ISO 286-1 dimensional size ranges from 1 mm to 18 mm in the X, Y, and Z directions. It was also driven by the need to reproduce the part several times and, by combining two parts, to ensure measurements in all ISO 286-1 dimensional size ranges from 18 mm to 180 mm. The main guidelines of ISO 52902 [18] were followed. Each

of the parts exhibits maximal dimensions of 18 mm × 18 mm × 18 mm across its X, Y, and Z axes, respectively. It is composed of several parallelepipeds arranged on top of a 5 mm thick base, preventing warping from occurring [20]. The design is symmetrical (Figure 1b). Fillets of 1 mm for the vertical edges and 0.500 mm for the horizontal edges (Figure 1a) complete the design and allow residual stresses to be reduced [29].

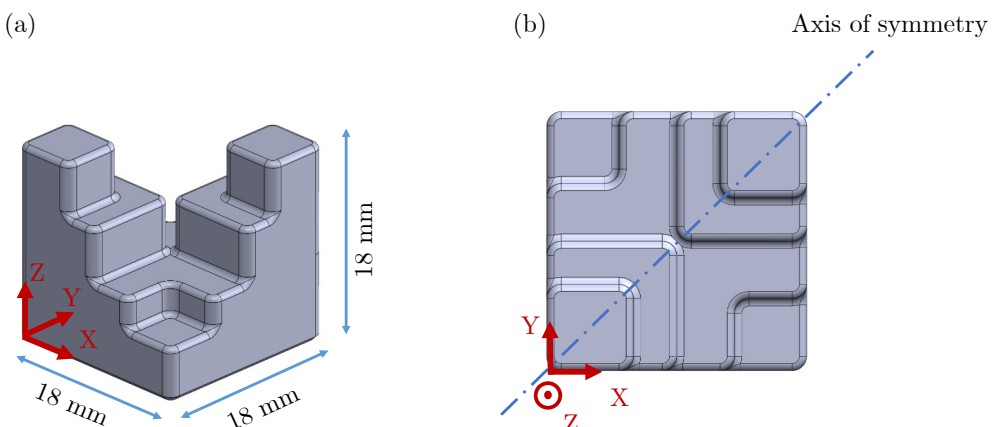

**Figure 1.** Design of the COMPAQT part with maximal dimensions (**a**) and axis of symmetry (**b**).

All parallelepiped dimensions were chosen to deliver at least 2 measurements for each dimensional size range of ISO 286-1. The dimensional measurements available in each part is given in Table 1 following the ISO 286-1 dimensional size ranges (in mm).

**Table 1.** Number of available measurements on a COMPAQT part for the dimensional size ranges of ISO 286-1.

| Size (mm) | X | Y | Z |
|---|---|---|---|
| 1–3 | 8 | 8 | 2 |
| 3–6 | 10 | 10 | 4 |
| 6–10 | 4 | 4 | 4 |
| 10–18 | 5 | 5 | 2 |

As the dimensions of the parts were limited to a maximum 18 mm for the X, Y, and Z axes, it is not possible to retrieve the machine performance for dimensions above 18 mm using a unique part. Combining the measurements between several parts replicated in the same batch at different places can solve the problem for the X and Y axes, while increasing the achievable sizes for the Z axis would require another design. However, the distribution of the parts depends on the printer to be tested and its available printing area.

### 2.2. Evaluated Machine

A MEX printer, Ultimaker 2+, was selected to conduct the machine performance analysis. This machine exhibited a printing volume of 223 mm × 223 mm × 205 mm (X, Y and Z) and is fitted with a Cartesian architecture (Z axis is the building direction). This machine can be used with several standard filaments such as ABS (Acrylonitrile Butadiene Styrene) or PLA (Polylactic Acid). PLA was chosen to conduct the experiments and print the required parts. It is a widespread material easy to print with lower health risk [22]. The filament roll supplier was Ultimaker, while its diameter was 2.850 mm. The CAD model was generated using SolidWorks 2021, while the slicing was performed using Cura 4.12.1. A .STL file was used between both software. It was exported from SolidWorks with the "fine" STL export preset, allowing a chord and angle deviation of 0.014 mm and 10° to be obtained, respectively.

To test as much as possible the available space on the build platform, 25 COMPAQT parts were evenly distributed over its surface, resulting in a 5 by 5 matrix of parts. This number of parts allows a significant number of measurements to be obtained for conducting a machine performance analysis [24]. Each part is located at 16 mm in X and Y from its nearest neighbor, as shown in Figure 2. This allows sizes of up to 154 mm to be evaluated, while ensuring at least 50 measurements to be obtained for each dimensional size range. The highest dimensional size range of ISO 286-1, which can be tested with the proposed parts' distribution, is 120–180 mm for the X and Y axes. In the case of the Z axis, the maximal size range that can be tested is 10–18 mm. As explained before, covering more sizes along the Z direction would require another part design, but it would increase the print time and required material. Moreover, MEX printers produce parts with high anisotropy when comparing the in-plane direction (X and Y axes) with the build direction (Z axis) [29]. The mechanical properties are lower along the build direction than in the in-plane direction. Therefore, the design was limited along the Z axis, while the replication of parts along the X and Y axes allow larger sizes in these directions to be evaluated.

The resulting printing area coverage by the COMPAQT parts was 54%. It is lower than the recommended 80% of coverage recommended by ISO 52902 for GBTA. However, the printing of parts covering more than the half of the printing area is not usual since the risk of warping and deformation of a part increases with the section length of the part in the X and Y planes of the build platform [29]. Moreover, the printing area coverage is slightly larger than that obtained from the study relying on 25 GBTAs for assessing the machine performance of Ultimaker 2+ (51%) [25].

Table 2 gives the total number of measurements that can be obtained with respect to the dimensional size ranges of ISO 286-1. As can be seen, two sets of data are available: the measurements specific to one part ("Part Specific" in the table) and the measurements using two parts ("Multi Part" in the table). In total, more than 1375 measurements were available both for the X and Y axes, while the number of measurements for the Z axis was 300.

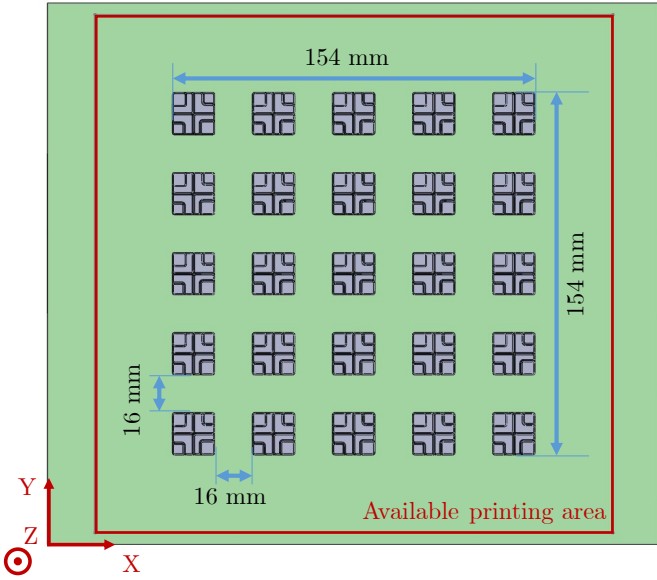

**Figure 2.** Distribution of the 25 COMPAQT parts on the build platform (in green) within the available printing area (red square).

**Table 2.** Number of available measurements for a batch of 25 parts depending on the dimensional size ranges of ISO 286-1. "Part Specific" stands for the measurements involving a single COMPAQT part, while "Multi Part" involves two COMPAQT parts.

| Size (mm) | X Multi Part | X Part Specific | Y Multi Part | Y Part Specific | Z Part Specific |
|---|---|---|---|---|---|
| 1–3 | | 200 | | 150 | 50 |
| 3–6 | | 250 | | 250 | 100 |
| 6–10 | | 100 | | 150 | 100 |
| 10–18 | | 125 | | 125 | 50 |
| 18–30 | 200 | | 200 | | |
| 30–50 | 200 | | 200 | | |
| 50–80 | 150 | | 150 | | |
| 80–120 | 100 | | 100 | | |
| 120–180 | 50 | | 50 | | |

To prevent parts' detachment from the build platform before their measurement, a brim of 12 mm was added around each COMPAQT part. A lacquer based on thermoplastic copolymers (3DLAC Original from 3D LAC) was also sprayed on the build platform before the print to enhance part adhesion. The relevant printing parameters are given in Table 3. The accuracy of the printed parts directly depends on the printing parameters' choice (e.g., layer thickness) and nozzle diameter [29]. For the proposed study, the printing parameters are derived from the filament supplier recommendations. This is why a layer thickness of 0.1 mm was selected. The printing time required to reproduce the 25 parts was 13 h. The as-built parts after printing are depicted in Figure 3. No post-treatment was applied to the parts after printing.

**Table 3.** Selected parameters to manufacture the parts.

| Parameters | Value |
|---|---|
| Nozzle diameter (mm) | 0.4 |
| Layer thickness (mm) | 0.1 |
| Density of infill (%) | 20 |
| Type of infill | Cubic |
| Build platform temperature (°C) | 60 |
| Nozzle temperature (°C) | 220 |
| Flow rate (%) | 100 |
| Printing speed (mm/s) | 50 |

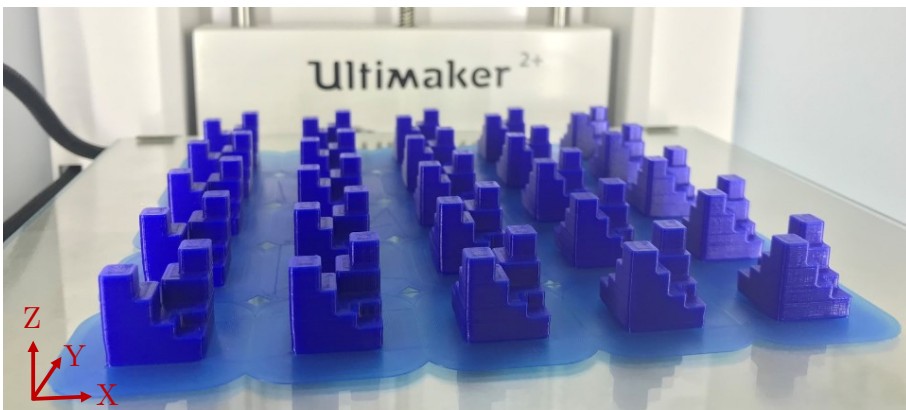

**Figure 3.** Batch of 25 COMPAQT parts after their manufacturing in an Ultimaker 2+ printer.

### 2.3. Measurement of Parts

The parts were measured using a Wenzel LH54 Coordinate Measuring Machine (CMM) with a spherical probe of 1.500 mm from Renishaw. The probe diameter allowed

the staircase effect of the parts to be filtered. For a given distance L (in mm) to evaluate, this machine exhibits a measurement uncertainty (in μm) of $3 + L/300$ for its X and Y axes and $3.5 + L/300$ for its Z axis.

All the parts were measured at room temperature directly on the glass build platform without being separated from it. The latter is easily removable from the printer, as for many of the available machines on the market. The measurement configuration is depicted in Figure 4. Measuring the parts directly on the build platform allows the measurements between two parts (Multi Part in Table 2) to be conducted. The parts' planes were evaluated by taking in general 6 points. The larger planes (composing the side of the parts) were measured using 10 points to better cover their size. However, for some planes with limited dimensions, only 4 points were accessible with the selected probe diameter. This is in accordance, nevertheless, with the minimum number of points recommended in the literature [30].

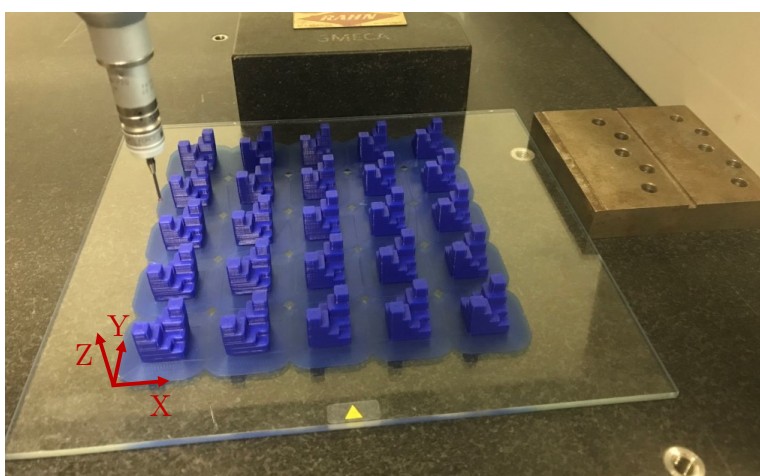

**Figure 4.** Measurement configuration used with the CMM.

The Metrosoft QUARTIS software version 2021 was used to pilot the CMM and obtain the several distances required for this study. This software computed the distance between the first plane and the second plane reduced as a point. The time required to measure all planes of the 25 parts was about 2 h, once the measuring program has been developed.

*2.4. Data Treatment and Indicators' Computation*

The collected measurements were post-treated following the ISO 22514 standard method:

- The measurements were grouped according to their axis and dimensions (all the measurements between 1 mm and 3 mm depending on the X axis from one group, for example).
- The deviation between the nominal value and the measurement was computed. This deviation was then used in the rest of the computations. For example, a measurement of 4.192 mm with a nominal value of 4 mm exhibited a deviation of 0.192 mm. Working with deviations eased the comparisons of results between distances belonging to distant size ranges.
- The best distribution fitting the data was chosen according to a Kolmogorov–Smirnov test. A normal, log-normal, folded-normal, gamma, Rayleigh, or Weibull distribution was then selected based on the test *p*-value, as recommended by ISO 22514.
- Using the fitted distribution, the necessary percentiles ($X_{0.135\%}$, $X_{50\%}$, and $X_{99.865\%}$) were computed. Finally, the potential and real machine performance indices $P_m$ and $P_{mk}$ were obtained following Equations (1) and (2) from ISO 22514. L and U stand for the lower and upper specification limits that have to be tested, respectively. In this

study, the Standard Tolerance Grades of ISO 286-1 (STGs in the rest of this paper) were used to obtain the lower and upper specification limits.

$$P_m = \frac{U - L}{X_{99.865\%} - X_{0.135\%}} \tag{1}$$

$$P_{mk} = min(\frac{X_{50\%} - L}{X_{50\%} - X_{0.135\%}}; \frac{U - X_{50\%}}{X_{99.865\%} - X_{50\%}}) \tag{2}$$

- From Equations (1) and (2), the achievable potential (Equation (3)) and real (Equation (4)) specification limits can be retrieved [22,31] for a given machine performance index. A reference value ensuring less than 1 out of a specification part for 1 million produced is 1.67 [24,26].

$$1.67 \times (X_{99.865\%} - X_{0.135\%}) = U - L \tag{3}$$

$$\begin{cases} L = X_{50\%} - 1.67 \times (X_{50\%} - X_{0.135\%}) \\ U = X_{50\%} + 1.67 \times (X_{99.865\%} - X_{50\%}) \end{cases} \tag{4}$$

- Finally, the uncertainty of the indices' estimation was evaluated by taking a 95% confidence interval and using the method of Bissel [32] and Chou et al. [33]. The minimal and maximal values for the $P_m$ and $P_{mk}$ indices can be computed through Equation (5) relying on a $\chi^2$ distribution. $n$ stands for the number of available measurements, while the confidence level is given by $1 - \alpha = 95\%$. This value is a standard in the literature [32–34]. Using these values of minimum and maximum for $P_m$ and $P_{mk}$ also allowed the uncertainty for the specification limit to be determined.

$$P_{m,min} = \sqrt{\frac{\chi^2_{n-1;\alpha/2}}{n-1}} \cdot P_m \leq P_m \leq \sqrt{\frac{\chi^2_{n-1;1-\alpha/2}}{n-1}} \cdot P_m = P_{m,max} \tag{5}$$

## 3. Results

### 3.1. Distribution Identification

As demonstrated in a recent publication [25], the best-fitting distribution is not always the normal. For example, Figure 5 gives the best-fitting distribution for the measurements belonging to the Y axis for the 50–80 mm ISO 286-1 class (150 measurements). As can be seen, the obtained gamma distribution very well fitted the dataset. The computed percentiles in this case were $X_{0.135\%}$ = −0.091 mm, $X_{50\%}$ = 0.152 mm, and $X_{99.865\%}$ = 0.494 mm. The *p*-value obtained for the gamma distribution was the highest among the tested distributions with a value of 0.997, far above the significance threshold of 0.05. The distribution was shifted to the positive deviations with a median of 0.152 mm. This shows a systematic error of the printer for those measurements.

The data set obtained from the measurements of the COMPAQT parts can be split into two different populations. Indeed, the measurements below 18 mm are part specific (Part Specific in Table 2), while the others above 18 mm are relying on two separated parts (Multi Part in Table 2). Therefore, the study of the best-fitting distribution was separated according to whether the measurement has the Part Specific character or not. Figures 6 and 7 depict the results' breakdown for the Part Specific and Multi Part measurements, respectively.

As can be seen in Figure 6, three distributions best fitted the Part Specific data. A total of 53% of the measurements were best fitted by a normal distribution, while 40% needed a gamma distribution. The remaining measurements (7%) were best fitted by a log-normal distribution.

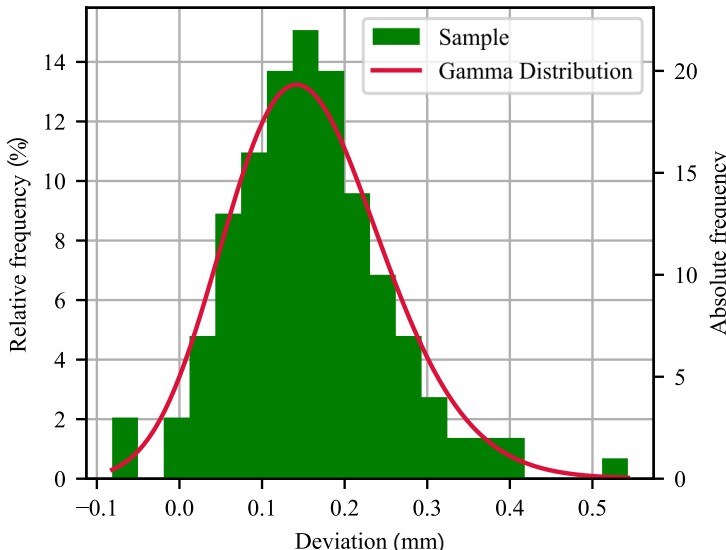

**Figure 5.** Example of fitting with a gamma distribution of the 150 measurements across the Y axis belonging to the 50–80 mm class.

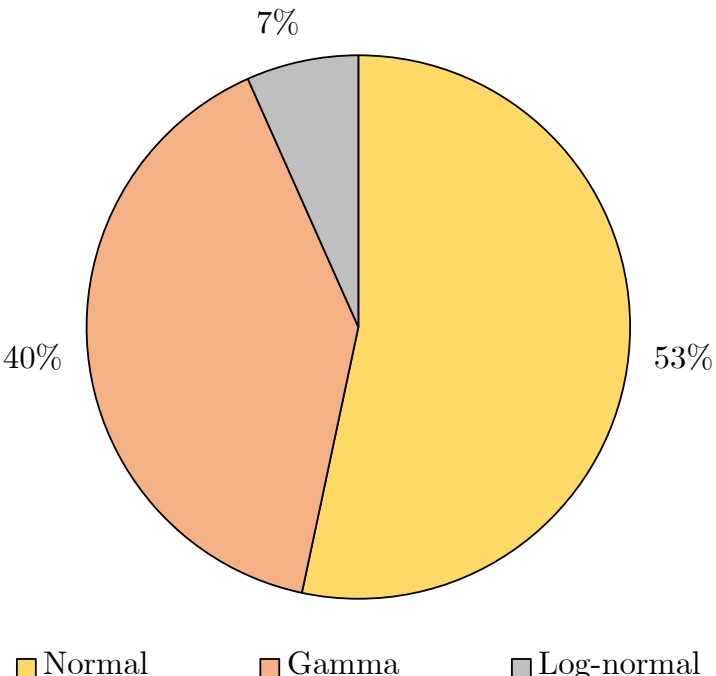

**Figure 6.** Best-fitting distributions for the Part Specific measurements (belonging to size ranges from 1–3 mm to 10–18 mm for the X, Y, and Z axes).

For the sake of Multi Part measurements, other distributions were needed. The normal distribution is still one of the best-fitting distributions for 30% of the measurements. However, another 30% of the measurements were best approached by a log-normal distribution, while the gamma distribution completed the top 3 by best-fitting 20% of the measurements. The Weibull and folded normal distributions were also required to fit 10% each of the measurements.

The different distributions needed to best fit the two different populations of data (Part Specific and Multi Part) can find their origin in the difference of thermal history between the parts. Indeed, depending on their positions in the build platform, they can endure

different cooling conditions, leading to different deformations. This is the first cause of inaccuracies of printed PLA parts according to Dantan et al. [35]. Moreover, in some cases, the resulting *p*-values of the different tested distributions were very close. This is expected since some distributions can approximate others: the Weibull distribution can approximate a normal distribution, for example [36]. As a result, some data sets were approximated with a Weibull distribution but were very close to a normal distribution. If the threshold of significance for the *p*-value of the statistical test was applied strictly ($p$-value $> 0.05$), it would mean that both distributions could have been chosen. However, the choice criterion was the highest *p*-value to ensure that a systematic decision would be taken.

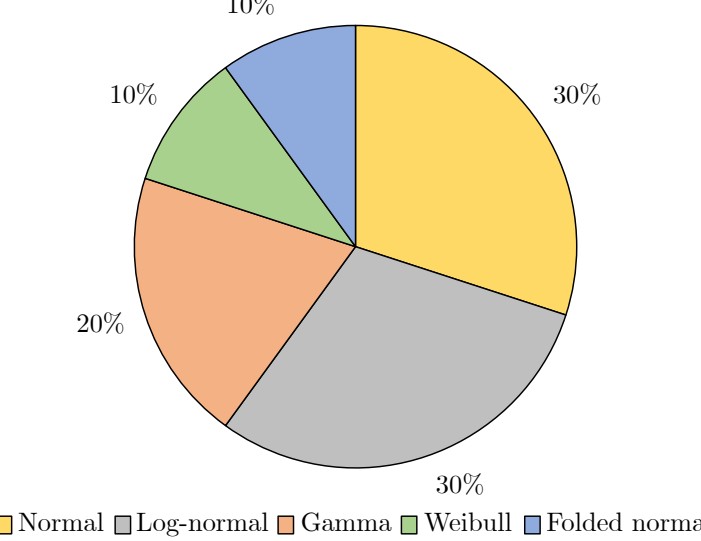

**Figure 7.** Best-fitting distributions for the Multi Part measurements (belonging to size ranges from 18–30 mm to 120–180 mm for the X, Y, and Z axes).

### 3.2. Potential Dimensional Tolerance Interval Capabilities

The potential dimensional tolerance interval capabilities (TICs) computed with a $P_m$ target of 1.67 are depicted in Figure 8. These are the TICs that can be potentially achieved by the printer if its systematic errors are compensated (at the design stage of the part to be printed, e.g., [37]). The dimensional size ranges of ISO 286-1 are depicted in the X axis of the graph. For each dimensional size range, two or three bars give the achievable potential TIC in mm. From 1 mm to 18 mm, the TICs displayed are related to single parts, while the TICs given for higher-dimensional size ranges involved two different parts. The error bars were determined to ensure an estimation of $P_m$ with a 95% confidence level.

As can be seen, independently of the dimensional size range, the X axis of the printer achieved lower TIC than its Y axis (except for the distances of planes between 10 mm and 18 mm). Since, the X and Y axes' design and motion transmission are identical, this suggests that the machine may require maintenance (e.g., extra lubrication, belt check). Indeed, the proposed method allowed for assessing the machine performance at a precise moment, while its performance can change over time (due to wear, for example). In this case, the machine was already used for several years. In an industrial context, the method can be applied at the reception of the machine and then at different times to verify if the machine is still able to meet its initial performances or if it requires extra maintenance. In the case of the Z axis, the performances are better than for the X and Y axes, whatever the considered dimension is. However, due to the design of the part, fewer measurements are available and allowed to assess the printer performances for maximal dimensions of up to 18 mm. The evaluation of the printer performances for higher Z distances would require another part design.

Following ISO 286-1, the higher the dimension, the higher the related TIC. However, as depicted in the graph, except for the Z axis measurements, the X and Y axes' results were

not strictly increasing. This can originate from the design of the used features [25]. Indeed, the design of the feature has a non-negligible influence on the evaluated performance. Since the planes composing each of the part are different, this may influence the results. Another factor that can influence the results is the difference in construction between the X–Y axes and the Z axis. Indeed, the linear joints composing the Z axis of Ultimaker 2+ are different and with higher rods' diameter. This can lead to a better rigidity and precision of movement along this axis. The X and Y axes' linear joints, on the other hand, exhibit a much lighter construction. Indeed, for a given layer, these two axes are moving the print head, while the Z axis remains fixed.

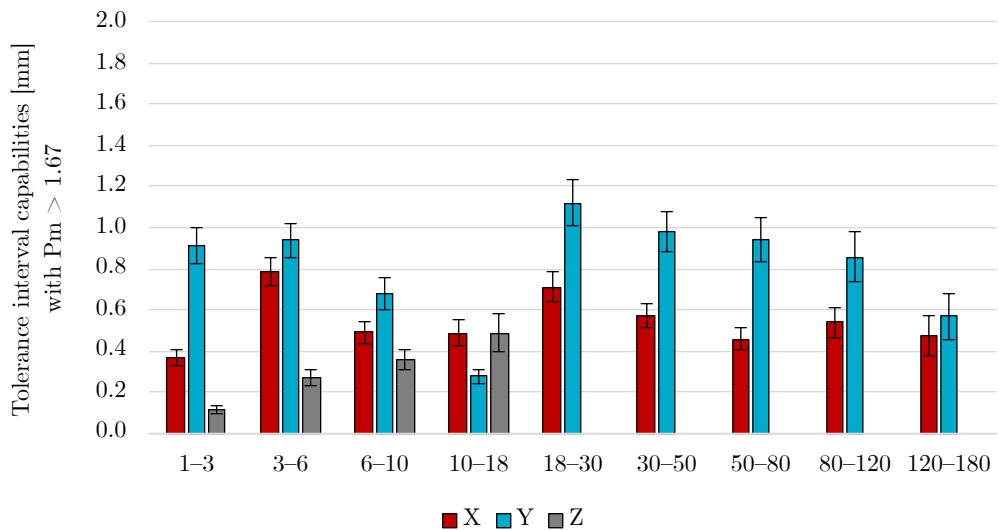

**Figure 8.** Potential tolerance interval capabilities in mm depending on the dimensional size ranges of ISO 286-1 considering a $P_m$ target of 1.67 at a confidence level of 95%.

Except for the Z axis, large TICs were achieved by the printer (up to nearly 1.200 mm for distances between 18 mm and 30 mm across the printer Y axis, for example). This is mainly due to the $P_m$ target used for this study. Indeed, the target $P_m = 1.67$ can be challenged depending on the foreseen application for the parts [25]. If a larger number of non-conforming parts can be produced, then a lower $P_m$ target can be used as $P_m = 1$. In this case, if the measurements were represented by a normal distribution, the number of non-conforming parts reaches 2600 ppm (parts per million), while it only reaches 0.54 ppm if $P_m = 1.67$ [24].

The same study was conducted with a $P_m$ target of 1, and the results are depicted in Figure 9. As can be seen, the achievable potential TICs all decreased by 59.9%. Indeed, changing the $P_m$ target from 1.67 to 1 in Equation (3) is the same as decreasing all the computed tolerance interval capabilities of 59.9% (1/1.67 = 0.599). For example, the TICs for dimensions across the printer's Y axis from 18 mm to 30 mm was reduced from about 1.200 mm to about 0.720 mm.

### 3.3. Real Dimensional Tolerance Interval Capabilities

The previous graphs showed the potential achievable TICs for the printer. However, these are the performances that can be achieved by the printer if its systematic errors were compensated. To obtain the real performances of the printer at the time of the test, the computation of $P_{mk}$ is needed. This allowed for obtaining the real TICs shown in Table 4. For each of the dimensional size ranges of ISO 286-1, the lower bound (L), middle bound (M), and upper bound (U) of the TICs are given in mm.

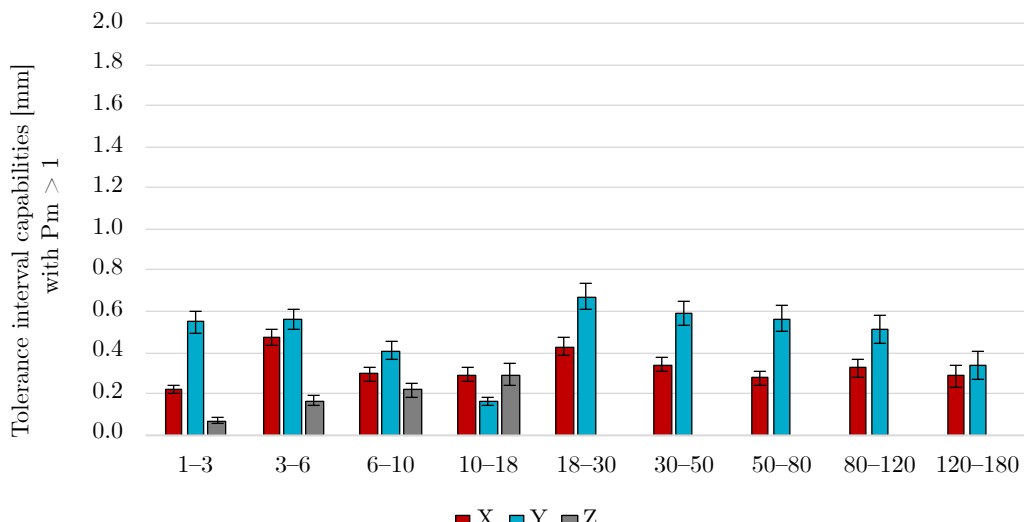

**Figure 9.** Potential tolerance interval capabilities in mm depending on the dimensional size ranges of ISO 286-1 considering a $P_m$ target of 1 at a confidence level of 95%.

As before, since the $P_{mk}$ target is set at 1.67, the resulting real TICs are large. For the X and Y axes and dimensions lower than 18 mm, most of the TICs exhibit a middle located around $0 \pm 0.050$ mm of deviation. Above 18 mm, there is a systematic error appearing and a higher shift of the middle of the TICs with values of up to 0.314 mm. Again, even with this shifting appearing, the X axis performances were better than those of the Y axis. In the case of the Z distances, the middle of the TICs is centered around $0 \pm 0.050$ mm of deviation as for the distances between 1 mm and 18 mm of the X and Y axes.

All distances above 18 mm for the X and Y axes are composed of measurements between two parts instead of one. This increases the risk of deviations since the deviations of two parts are added. According to Dantan et al., the three highest contributors to the inaccuracies of printed PLA parts are the cooling-down process endured by the parts after their fabrication (and their resulting shrinkage), inaccuracies from the machine movements' control, and the mechanical imperfections of the machine itself [35]. The management of the cooling-down process was the contributor with the highest impact on the inaccuracy of the parts printed by this research team. To best capture the distortions caused by the non-homogeneous cooling down of the part, simulations are required [38]. Preventing the inaccuracies and avoiding high temperature gradients by tuning the part's cooling down could lead to better performances of the printing. However, this is not yet taken into account in widespread slicers such as the one selected for this study (Cura version 4.12.1).

**Table 4.** Real tolerance interval capabilities computed considering a $P_{mk}$ target of 1.67 at a confidence level of 95%, lower bound (L), middle bound (M), and upper bound (U) of the interval, all in mm.

| Size (mm) | X | | | Y | | | Z | | |
|---|---|---|---|---|---|---|---|---|---|
| | L (mm) | M (mm) | U (mm) | L (mm) | M (mm) | U (mm) | L (mm) | M (mm) | U (mm) |
| 1–3 | −0.137 | 0.046 | 0.228 | −0.502 | −0.035 | 0.432 | −0.013 | 0.047 | 0.107 |
| 3–6 | −0.387 | 0.017 | 0.421 | −0.493 | −0.013 | 0.467 | −0.150 | −0.010 | 0.129 |
| 6–10 | −0.262 | −0.011 | 0.241 | −0.361 | −0.011 | 0.339 | −0.205 | −0.021 | 0.163 |
| 10–18 | −0.240 | 0.012 | 0.264 | −0.159 | −0.022 | 0.116 | −0.236 | 0.018 | 0.272 |
| 18–30 | −0.286 | 0.081 | 0.448 | −0.465 | 0.107 | 0.678 | | | |
| 30–50 | −0.227 | 0.067 | 0.361 | −0.409 | 0.093 | 0.595 | | | |
| 50–80 | −0.085 | 0.153 | 0.391 | −0.256 | 0.227 | 0.711 | | | |
| 80–120 | −0.183 | 0.098 | 0.379 | −0.124 | 0.314 | 0.751 | | | |
| 120–180 | 0.035 | 0.282 | 0.529 | 0.009 | 0.304 | 0.599 | | | |

Again, depending on the foreseen application for the parts, the $P_{mk}$ target can be challenged and reduced from 1.67 to 1.0. The results by considering this target are given in Table 5 with the lower bound (L), middle bound (M), and upper bound (U) of the real TICs in mm. Again, this change of $P_{mk}$ target leads to narrower TICs, while their middle position and the resulting shifting are unchanged. Compensating the shifting of the TICs' middle at the design stage can be a solution to recenter the TICs at 0 mm. This is of prime interest and can be a perspective of this work.

**Table 5.** Real tolerance interval capabilities computed considering a $P_{mk}$ target of 1 at a confidence level of 95%, lower bound (L), middle bound (M), and upper bound (U) of the interval, all in mm.

| Size (mm) | X | | | Y | | | Z | | |
|---|---|---|---|---|---|---|---|---|---|
| | L (mm) | M (mm) | U (mm) | L (mm) | M (mm) | U (mm) | L (mm) | M (mm) | U (mm) |
| 1–3 | −0.088 | 0.022 | 0.131 | −0.314 | −0.035 | 0.245 | −0.003 | 0.033 | 0.068 |
| 3–6 | −0.232 | 0.010 | 0.252 | −0.300 | −0.013 | 0.275 | −0.094 | −0.011 | 0.073 |
| 6–10 | −0.161 | −0.011 | 0.140 | −0.221 | −0.012 | 0.198 | −0.131 | −0.021 | 0.089 |
| 10–18 | −0.153 | −0.002 | 0.149 | −0.107 | −0.024 | 0.058 | −0.145 | 0.008 | 0.160 |
| 18–30 | −0.139 | 0.081 | 0.301 | −0.239 | 0.104 | 0.446 | | | |
| 30–50 | −0.112 | 0.064 | 0.240 | −0.207 | 0.094 | 0.394 | | | |
| 50–80 | −0.001 | 0.142 | 0.284 | −0.092 | 0.198 | 0.487 | | | |
| 80–120 | −0.030 | 0.139 | 0.307 | 0.010 | 0.272 | 0.534 | | | |
| 120–180 | 0.125 | 0.273 | 0.421 | 0.127 | 0.304 | 0.481 | | | |

### 3.4. Potential Achievable Standard Tolerance Grades of ISO 286-1

Since Additive Manufacturing is still a young process, it lacks standards, especially when tolerancing the produced parts is required. For that purpose, existing standards, initially developed for conventional processes such as machining, can be used according to ISO 17296-3 [15]. ISO 286-1, for example, gives Standard Tolerance Grades (STGs) from 1 to 18 for dimensions of up to 3150 mm. The lower the STGs, the narrower the associated tolerance interval capabilities.

These STGs were used to evaluate the dimensional performances of the printer depending on the different size ranges. Table 6 gives the potential achievable STG for each axis and dimensional size range taking a $P_m$ target of 1.67 as an objective. As can be seen in the table, the X axis delivered better results (tighter tolerance interval capabilities) than the Y axis for almost all dimensional size ranges, except for dimensions between 10 mm and 18 mm where the Y axis exhibited a better performance. This is in accordance with the potential tolerance interval capabilities presented previously. Moreover, the Z axis showed the same behavior as before with lower achievable STGs than the X and Y axes. Nevertheless, the overall STGs achieved were far above the those of conventional processes such as milling. Indeed, milling process with carbide tools can achieve commonly a STG of 7, while it can reach lower levels as 4 or 5 in specific conditions [39]. This shows, again, all the potential brought by hybrid machines combining additive and subtractive processes. Indeed, the production and finishing of the part can be directly foreseen in the same machine, while the tight tolerances and smooth surface topographies of milling can be achieved for an additively manufactured part after its fabrication.

Nevertheless, working with STGs is not as precise as computing the real tolerance interval capabilities as before. Indeed, STGs are classes. For example, for the distances between 80 mm and 120 mm of the Y axis, the $P_m$ index was 1.660 for STG 14 and 2.652 for STG 15. Since the $P_m$ target was 1.67, STG 15 was selected even if the $P_m$ of STG 14 was very close to the target of 1.67. Therefore, precautions should be taken when assessing the machine performance of a printer based on classes such as the STGs.

**Table 6.** Potential achievable Standard Tolerance Grades of ISO 286-1 at a confidence level of 95%.

| Size (mm) | X | Y | Z |
|---|---|---|---|
| 1–3 | 15 | 17 | 13 |
| 3–6 | 17 | 17 | 14 |
| 6–10 | 15 | 16 | 15 |
| 10–18 | 15 | 14 | 15 |
| 18–30 | 15 | 16 | |
| 30–50 | 14 | 16 | |
| 50–80 | 14 | 15 | |
| 80–120 | 14 | 15 | |
| 120–180 | 13 | 13 | |

*3.5. Real Achievable STGs of ISO 286-1*

The STGs presented in Table 6 are the potential achievable and not the real ones. Indeed, as before, this approach allows us to see the potential STGs that can be achieved by the printer if there was no systematic error. Again, using $P_{mk}$ instead of $P_m$ allowed us to take into account the systematic errors. The real STGs achievable by the printer were then computed considering a $P_{mk}$ target of 1.67 and a confidence level of 95%. The results are given in Table 7. As before, the STGs are given for each axis and dimensional size range. For each axis, three columns give the potential (Pot.) and real (Real) STGs and their difference (Diff.). A color code shows if there is a difference of one STG (orange), two STGs (red), or no difference (green).

As exhibited previously with potential and real tolerance interval capabilities, there were differences for some dimensional size ranges. Most differences were of one STG between the potential and real STGs, while the highest dimensional size ranges (120 mm to 180 mm) exhibited the highest difference with two STGs. These differences are in line with the occurrence of a shifting of distribution as exhibited by a middle of the tolerance interval (M in Table 4) not close to zero. Therefore, the higher the shifting of distribution in Table 4, the higher the difference between the potential and real STGs. As a result, the dimensional size ranges of 120 mm to 180 mm for the X and Y axes were those with the highest difference between the potential and real STGs. Indeed, the dimensions encompassed in these ranges exhibited among the highest shiftings of distribution, as shown in Table 4, with values of 0.282 mm and 0.304 mm for the X and Y axes, respectively. The dimensions belonging to the Y axis between 50 mm and 80 mm and between 80 mm and 120 mm exhibited also among the highest deviations. However, their tolerance interval middle was less shifted with respect to the target of 0 mm. Indeed, their lower bounds (L in Table 4) were −0.256 mm and −0.124 mm, respectively. By contrast, the dimensions belonging to 120 mm to 180 mm exhibited a larger lower bound of 0.009 mm.

Again, great care should be taken with the achievable STGs since they are classes and not absolute values of tolerance interval capabilities. Anyway, the STGs allow an easier comparison with the performances of conventional processes such as milling or injection molding.

*3.6. Influence of Temperature on the Parts' Accuracy*

The build volume of Ultimaker 2+ was closed during the print by a door and the top of the printer by a cover. However, no active control of temperature was available on the printer. Therefore, the temperature within the build volume could vary over the printing time. This could affect the parts' accuracy since, at different locations on the build platform, they might exhibit different cooling rates and, therefore, different deformations. To better observe the influence of the parts' positions within the build volume, an indicator was built averaging the relative deviations measured on the parts along the X, Y, and Z axes. The Part Specific measurements were used, from 1 mm to 18 mm. This relative deviation is a dimensionless number obtained by dividing the deviation by its nominal

size. For example, a measured deviation of −0.037 mm for a nominal size of 25 mm would give a −0.15% relative deviation.

**Table 7.** Potential (Pot.) and real (Real) achievable STGs of ISO 286-1 at a confidence level of 95% and their difference (Diff.). Green is used for cells without a difference between potential and real STGs, orange for the difference of one STG, and red for a difference of two STGs.

| Size (mm) | X Pot. | X Real | X Diff. | Y Pot. | Y Real | Y Diff. | Z Pot. | Z Real | Z Diff. |
|---|---|---|---|---|---|---|---|---|---|
| 1–3 | 15 | 16 | 1 | 17 | 17 | 0 | 13 | 14 | 1 |
| 3–6 | 17 | 17 | 0 | 17 | 17 | 0 | 14 | 14 | 0 |
| 6–10 | 15 | 15 | 0 | 16 | 16 | 0 | 15 | 15 | 0 |
| 10–18 | 15 | 15 | 0 | 14 | 14 | 0 | 15 | 15 | 0 |
| 18–30 | 15 | 16 | 1 | 16 | 17 | 1 | | | |
| 30–50 | 14 | 15 | 1 | 16 | 16 | 0 | | | |
| 50–80 | 14 | 15 | 1 | 15 | 16 | 1 | | | |
| 80–120 | 14 | 14 | 0 | 15 | 16 | 1 | | | |
| 120–180 | 13 | 15 | 2 | 13 | 15 | 2 | | | |

Figure 10 gives the results of this analysis for the distances measured along the X, Y, and Z axes and a global indicator averaging the results of all axes. Each sub-table represents the build platform with the results of each of the 25 parts. The rear of the printer and its door are indicated. A color code was added to better see the highest and lowest values within each size range. Blue is used for negative relative deviations, red for the positive, and white for the values approaching zero. As depicted in the picture, no direct tendency can be retrieved from the table, except the fact that, depending on the location on the build platform, different values of deviations were observed. The parts located in the vicinity of the door did not exhibit much higher deviations compared with those located at the center of the build platform or at the printer's back. In the same way, the parts located at the center of the build platform did not exhibit lower deviations.

The absence of tendency in the results comes from the use of offline metrology and the measurement of printed parts. Indeed, artifacts such as COMPAQT allow for evaluating multiple errors that combine when producing a part by AM [40]. Therefore, the temperature gradient inside the build volume could explain some of the observed deviations, but other factors can also contribute to them. These can be, for example, the accumulation of errors of the axes' encoders over the print, the slippage of the filament inside the filament feeder, the path planning of the nozzle to generate the different parts on the build platform, or the deflection of the linear guiding depending on the nozzle locations.

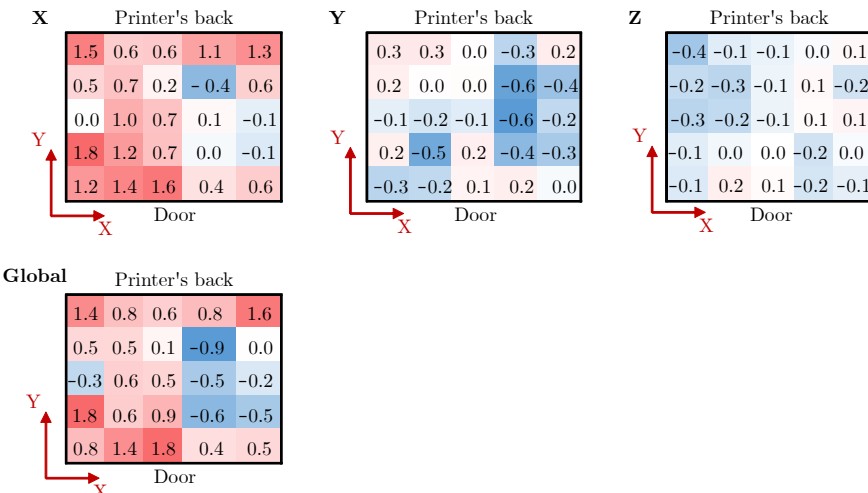

**Figure 10.** Average relative deviations in % depending on the X, Y, and Z axes and global indicator, blue is used for negative relative deviations, red for the positive, and white for the values approaching zero.

## 4. Discussion

Compared with the study using GBTA to assess the machine performance of MEX printers [25], the proposed method using COMPAQT parts is 98% faster. Indeed, the required printing time to obtain the 25 COMPAQT parts was 13 h, while the method based on 25 GBTAs required 675 h. Moreover, the proposed method allowed the same systematic assessment of dimensional performances to be conducted, while covering all the available dimensional size ranges of ISO 286-1 from 1 mm to 180 mm for the X and Y axes. However, the method using the 25 GBTAs provided also geometrical measurements, which were not evaluated with the COMPAQT parts. Indeed, the several planes composing the COMPAQT part design do not allow for systematically covering all the achievable size ranges of ISO 2768-2 for the flatness, perpendicularity, and parallelism.

The replication of the 25 COMPAQT parts is able to give measurements of up to the range of 120 mm to 180 mm of ISO 286-1, while the GBTA provided measurements of up to the range of 180 mm to 250 mm. For the COMPAQT parts, the choice was made to align with the conventional use of desktop MEX printers. Indeed, most of the users print parts located at the center of the printer and avoid approaching the edges of the build platform since the printer's performances decrease at these extreme locations. Nevertheless, if knowing the performances of the printer up to the highest achievable size range was necessary, more COMPAQT parts could be printed. Indeed, printing 36 parts in a 6 by 6 matrix (with 16 mm between the parts, as previously shown in Figure 2) instead of the 25 parts in 5 by 5 matrix provides measurements of up to 188 mm. The printing time in this case is foreseen to reach 18 h instead of 13 h (40% increase), while the mass of material needed to perform the study would increase from 56 g to 81 g (almost 45% increase). Limiting the number of parts to 25, therefore, allowed for rationalizing the time and material needed for obtaining the machine performance of the printer, while aligning with the users' habits.

Since the slicing is also a part of the machine data workflow, the CAD model exported to STL parameters could also have an influence on the obtained machine performance. The chosen parameters (preset "fine" in SolidWorks) with a chord deviation of 0.014 mm and an angle deviation of 10 degrees are following the recommendations that can be found in ISO 52902 when converting the CAD model to the STL file format. Indeed, this standard recommends have a chord deviation in the order of magnitude of a tenth of the expected accuracy of the AM system. The printer is depositing layers of 0.1 mm, so the criterion is fulfilled. However, further investigations could be conducted using the COMPAQT part to assess the influence of the STL conversion parameters and quantify it with the achievable tolerance interval capabilities.

Depending on the foreseen application, the method based on either the GBTA or the COMPAQT part can be recommended. Indeed, the method based on the GBTA is recommended for R&D purposes, which are less sensitive to the time required to produce the parts, while seeking the most complete picture of the printer's performances. In the case of an industrial production follow-up, the COMPAQT part method is best suited since it allowed for obtaining the same level of details as the GBTA for the dimensional performances, while bringing significant time savings. In addition, the volume of material required is smaller, which also reduced the feedstock costs.

The presented method can be extended to other printers by reproducing the COMPAQT part design at least 25 times on their build platform, keeping the same space between them as presented previously (16 mm along the X and Y axes). For larger printers, the number of parts can be increased to access and evaluate the features with dimensions larger than the 120 mm to 180 mm range. However, when testing this COMPAQT part on other printers, some care should be taken with the printing parameters' selection. Indeed, they must comply with the features of the design. Printing the COMPAQT part with a higher nozzle diameter of 6 mm instead of 0.4 mm, for example, will generate much higher layer heights (3 mm). In this case, it means that the vertical walls of the part with a 3 mm height will be done only in 1 layer instead of 30 with a 0.1 mm layer height. The wall generated will then exhibit a curved surface not relevant for the study. Moreover, some of the features

composing the COMPAQT part would not be possible to print. Indeed, the fillets on the vertical edges exhibit a radius of 1 mm, while for a MEX printer, the minimal achievable radius of curvature is equivalent to that of the nozzle [29]. Furthermore, the smallest vertical plane is 3 mm high, resulting in a 3 mm layer height, in a single layer with a non-flat surface. With respect to the features of the COMPAQT part, the proposed design could be printed with a maximal nozzle diameter of 2 mm with a 1 mm layer height. Deriving designs based on the same principle (stairs of different height) but with adapted features can be foreseen when evaluating printers using a layer height thicker than 1 mm.

## 5. Future Prospects

Assessing the influence of the printing parameters on the tolerance interval capabilities of the printer can be a perspective of this study. Indeed, the resulting accuracy of the printer depends of the printing parameters [41]. Reproducing the 25 parts with different sets of parameters can be used to highlight their influence on the machine performance and optimize them to minimize the printing time. An inspiring method can be found in the paper of Papazetis and Vosniakos to pursue this goal [42]. In addition, it was not possible to draw direct conclusions about the potential influence of the temperature in the build volume on the accuracy of the parts. Since an artifact such as COMPAQT measures a combination of several errors that affect its accuracy, further investigations could be conducted using, for example, a thermal camera and simulation tools.

Another output of the presented method can be the comparison of the performances of multiple identical printers to evaluate machine-to-machine variability [43]. This topic remains poorly covered in the literature, while companies operate several identical printers (from 5 to 500) to parallelize the production of parts, enhance the productivity, and decrease the costs [44]. Producing a set of 25 COMPAQT parts on several identical printers would allow for measuring the printer-to-printer variability, while covering the different size ranges of ISO 286-1 for the X and Y axes and those from 1 mm to 18 mm for the Z axis.

The COMPAQT part design can also be used to perform a quality and performance follow-up of a given machine. Indeed, this design can be incorporated into batches producing other parts. Measuring this part and comparing its dimensions with the tolerance interval capabilities assessed during the machine performance analysis allows a quality and performance follow-up to be conducted. This kind of monitoring is useful to foresee the machine maintenance and anticipate breakdowns. Finally, the method can also be used at the reception of a machine when commissioning it to verify if the tolerance interval capabilities announced by the supplier are achievable or not.

## 6. Conclusions

The proposed method and COMPAQT part design allow for assessing the machine performances of MEX printers for dimensional measurements, while keeping the parts' manufacturing time lower than 24 h. The method is suitable for production purposes and allows for performing a reception test (when commissioning a machine) or monitoring the longer-term production.

The main conclusions of this study are as follows:

- The proposed method allowed for determining the potential and real machine performances of the studied printer. The commonly used target of 1.67 was considered for $P_m$ and $P_{mk}$. Depending on the final application of the foreseen produced parts, it can be challenged and lowered to 1.
- Independently of the dimensional size range, the features aligned with the X axis achieved lower TICs than those aligned with the Y axis. The Z axis exhibited the best performance among the three axes, but no measurements were available for size ranges above the 10 mm to 18 mm category due to the design of the COMPAQT part.
- The Multi Part measurements exhibited a higher systematic error (up to 0.314 mm) than the Part Specific measurements (TICs centered around 0 mm $\pm$ 0.050 mm). This

may originate from inadequate cooling management of the parts in the gcode or from the machine's inaccuracies of movements or imperfections.

- The potential and real machine performances relying on the ISO 286-1 Standard Tolerance Grades (STGs) were determined. The X axis measurements reached, on average, one lower STG than the Y axis, confirming again the better performances across the X axis. The best STGs were achieved by the Z axis with a slightly better performance than the X axis. The potential and real STGs were slightly different for some size ranges. The higher the shifting of the TIC center, the higher the difference between the potential and real STGs.
- The time needed to produce the batch of the 25 parts (13 h) enables its wide spread in the industry for the reception tests of new machines or for the performance assessment of existing ones. Moreover, the compact design of the part allows its incorporation into batches of another part to perform a long-term quality follow-up.

**Author Contributions:** Conceptualization, F.D., E.R.-L., L.S., P.-J.A., V.D. and E.N.F.; methodology, L.S. and E.N.F.; software, L.S.; validation, F.D., E.R.-L., V.D. and L.S.; formal analysis, L.S.; investigation, L.S. and E.N.F.; resources, F.D. and E.R.-L.; data curation, L.S.; writing—original draft preparation, L.S.; writing—review and editing, F.D., E.R.-L., P.-J.A., V.D. and L.S.; visualization, L.S.; supervision, F.D., E.R.-L. and P.-J.A.; project administration, L.S., F.D., E.R.-L. and P.-J.A.; funding acquisition, F.D. and E.R.-L. All authors have read and agreed to the published version of the manuscript.

**Funding:** The European Union and the Walloon regional government funded this research (HybridAM research project).

**Data Availability Statement:** The data presented in this study are available on request from the corresponding author.

**Conflicts of Interest:** The authors declare no conflicts of interest.

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
