# Peer review of "Faster Evaluation of Dimensional Machine Performance in Additive Manufacturing by Using COMPAQT Parts"

_jmmp, doi:10.3390/jmmp8030100_

Round 1
Reviewer 1 Report
Comments and Suggestions for Authors
The proposed methodology introduces a novel approach to assess machine performance in AM, addressing the challenge of determining tolerances while minimizing manufacturing time. The method could contribute to advancing industry standards and facilitating research in the field. However, the manuscript could benefit from improving the phrasing, extending the discussion on the limitations of the proposed GBTA, and a larger experimental validation.
I recommend the authors to check the work of Lieneke et al.:
https://hdl.handle.net/2152/89334
I believe there is a second work from this group on the topic, but I cannot find a link.
One thing missing in the GBTA proposed by the authors is a broader range of geometries. No circularity, cylindricity, holes, flatness, etc. can be tested. Compared to other alternatives, this is a significant handicap.
When referring to the ability of a machine/process to produce parts within specified tolerances, it is more appropriate to use the term "tolerance capabilities" instead of "tolerance intervals." "Tolerance capabilities" refers to the consistency of a machine or process in producing parts that fall within specified tolerance limits. Another option could be "tolerance interval capabilities", which emphasizes the machine/process's ability to operate within predefined tolerance limits. This difference is not explicitly stated in the document.
The ISO 22514 standard states that it is not suitable for complex geometrical measurement processes, such as surface texture and position measurements that rely on several measurement points or simultaneous measurements in several directions. I wonder how this fits with the use of a CMM. I would like to see some clarification on this matter.
Further details of the equipment must be provided: It is not suitable for complex geometrical measurement processes, such as surface texture and position measurements that rely on several measurement points or simultaneous measurements in several directions.
Although 25 specimens were printed, the test was run just one time. Considering the significant reduction of time compared to previous methods, it makes sense to run the test multiple times and run a statistical analysis of the different printing tests.
Was there a reason to constrain the printing area to ~50%? The only justification provided was that a previous study by the same author did the same. Avoiding the worst-performing features of the machine adds undesired bias to the study. Also, see the next point.
The temperature on the printing bed/chamber is rarely uniform, which should lead to deviations in specimens when comparing specimens in different positions on the printing bed. Was that the case between the 25 samples?
Although I understand the rationales behind comparing different distributions (normal vs Weibull vs etc.) to assess the best option, I would like a closing argument on the matter. Which should be used? How large is the discrepancy when comparing a Weibull or Normal distribution? Is the difference significant? Also, please check the ongoing debate on the relevance of the p-value: https://doi.org/10.3344%2Fkjp.2017.30.4.241
The slicer version used (Cura 4.12.1) in processing should be listed in the materials and methods section. It's listed only in the Results section (line 338).
Because the slicer determines the path and how the outer shell is built (wipe, seams, etc.), the assessment process should probably not just consider the machine but the entire digital-to-solid process. From my experience, even the quality of the STL files could influence the outcome, although no study has been done so far. Furthermore, as you mention in lines 343-345, this deviation can be compensated by the software (implicitly) or explicitly through the slicer parameters (scaling the model in the X-Y direction).
While variations in the X and Y directions shown in Fig. 7 and 8 appear to be random, there's a clear positive trend in Z, with tolerances increasing almost linearly as the nominal dimension grows. This should be addressed (probably related to the lead-screw mechanism in the Z-axis). Could a change in the current COMPAQT address this issue?
Specific comments by lines:
Line 23-25: There are already many companies using AM for mass production. Prusa 3D printers is just one of several examples...
Line 28-30: Not all applications require a smooth surface finish or post-processing.
Lines 40-46: ISO 286, ISO 2768 are not tied to a specific manufacturing process. They are related to design/part specifications and can be applied to any part built using AM or any other process. The tolerance capabilities of a process, on the other hand, are used to determine if the process can achieve the desired tolerance set by the function and specified using those standards.
Lines 440-452: These are not evaluated in the current study. Hence, it does not belong in the Results section. It should be moved to future studies.
Comments on the Quality of English LanguagePlease check for grammar and English errors:
Line 285-286: The sentence needs clarification ("at a given time"?).
Line 287: Correct "the machine was already used for since several years"
Line 347-348: See comments above for Line 40-46.
Line 356: I do not understand what "more performant" means. Better performance? Tighter tolerances?
Line 428-430: The meaning of the sentence is unclear. Please rewrite.
Reviewer 2 Report
Comments and Suggestions for Authors
The article's authors undertook an interesting study on determining the dimensional accuracy of a sample made by the additive method. This procedure aimed to determine the performance of the 3D printer. Below are the essential comments and questions for the article:
Editorial Notes:
(a) I would ask in the abstract to include the statistical values obtained during the research presented (please clarify the abstract)
(b) from my perspective, creating in the introduction subsections 1.1 and 1.2. is unnecessary. The content of these subsections as much as possible is appropriate (very widely described), although it would be worthwhile to shorten at least the first two paragraphs.
(c) I would rename Chapter 3 as just Results, and replace subsection 3.6 with Chapter 4 - Discussion
(d) It seems to me that on the editing side of references were developed in accordance with the format. To be sure, I would ask you to verify them again
Substantive comments
(a) I am glad the authors have developed the publication quite thoroughly based on the study of the standard. What led to the conception of the model in Figure 1? From their own experience or based on the ISO/ASTM 52902 standard? In addition, do the authors plan to develop another model based on such a model that considers internal dimensions?
In addition, do the authors plan further research related to the design of models (such as in literature item No. 25) for testing 3D printers
(b) What were the parameters for exporting a 3D-CAD file to 3D-STL in SolidWorks? I am referring to the chord and angle deviation. Exporting errors could have affected the manufacturing errors.
(c) I don't know what material was used to make the samples. Was it ABS or PLA? Why was such a material chosen and not another? What motivated the authors to use such a thickness of the print layer? This parameter also has an impact (like a material) on the quality of the 3D printing.
(d) How were the models positioned and restrained to the worktable during measurement on the CMM? It would be helpful to post a photo demonstrating the measurement process.
(e) As I understand it, based on measurements of points on the surface of the models, planes were created based on which the measured dimensions were obtained. What computational methods (algorithms) were used to obtain the results of length measurements.
Comments on the Quality of English LanguageMinor editing of English language required
Round 2
Reviewer 2 Report
Comments and Suggestions for Authors
The article's authors responded to all the reviewer's comments and questions. After reviewing them and studying the article after revisions, the reviewer accepted the article in its current form.
Comments on the Quality of English LanguageMinor editing of English language required.